# The effect of health-promoting leadership of nursing managers on the work withdrawal behaviors and psychological safety of nurses

Darya Esmaeilbeigi[1], Mahdi Sahraei Beiranvand[2], Fatemeh Mohammadipour[3]*

1 Student Research Committee, Lorestan University of Medical Sciences, Khorramabad, Iran,
2 Department of Educational Sciences, Faculty of Education and Psychology, Shahid Beheshti University, Tehran, Iran, 3 School of Nursing and Midwifery, Social Determinants of Health Research Center, Lorestan University of Medical Sciences, Khorramabad, Iran

* mohamma.fafa@gmail.com

## Abstract

### Aim

Health-promoting leadership is a novel organizational strategy aimed at preventing the occurrence of diseases in the workplace and promoting the health needs of staff. The present study aimed to investigate the effect of health-promoting leadership of nursing managers on the nurses' withdrawal behaviors and psychological safety.

### Design

Structural equation modelling was used to verify the conceptual model.

### Methods

A total of 346 nurses working in educational hospitals in the western provinces of Iran were included using stratified random sampling by 2023. The research instruments included a checklist of demographic and occupational characteristics, health-promoting leadership, withdrawal behaviors, and psychological safety questionnaires. Pearson's correlation and structural equation modelling were used to investigate the relationships between the variables. We followed the STROBE Checklist.

### Results

The findings showed a significant negative relationship between health-promoting leadership and withdrawal behaviors and a significant positive relationship with nurses' psychological safety. There was also a negative and significant relationship between the withdrawal behaviors and psychological safety. Health-promoting leadership can explain 45% and 52% of the variance in psychological safety and withdrawal behaviors, respectively.

**Data availability statement:** Data cannot be shared publicly because of The ethical, confidential, and organizational nature of the variables under investigation. Data are available from the Iran National Committee for Ethics in Biomedical Research (contact via: ethics@beh-dasht.gov.ir, Phone umber: +982181455618) for researchers who meet the criteria for access to confidential data.

**Funding:** The author(s) received no specific funding for this work.

**Competing interests:** The authors have declared that no competing interests exist.

## Conclusions

While emphasizing the importance of leadership styles, the findings of this study indicate that nurse leaders should acknowledge the significance of health promotion leadership as a crucial set of behaviors to foster psychological safety and reduce withdrawal behaviors among nurses, thereby enhancing their quality of care and performance.

## Implications for nursing management

The results draw the attention of nursing managers to health-promoting leadership and its relationships, and create an impetus for changing leadership styles and addressing the lack of mutual understanding. Considering this finding, it appears necessary to create clearer prospect for the nursing profession by teaching health-promoting leadership principles to nursing managers and nurses without managerial positions.

## Introduction

Health is a critical factor that individuals seek to maintain, and employees desire work environments that promote their well-being [1]. In medical settings, occupational hazards, including extended working hours and chronic stress, negatively impact the health of healthcare personnel, with nurses, as essential members of the health system, being exposed to extensive physical, biological, chemical, psychological, and ergonomic risks [2,3]. Additionally, the recent Covid-19 pandemic has played a prominent role in exacerbating physical and psychological burdens [4]. The healthcare system that purports to protect public health lacks a comprehensive plan to ensure and maintain the health of its staff, which has consequently become a significant challenge [2]. It is evident that neglecting nurses' health affects healthcare system performance and diminishes quality of care [5]. In other words, leaders who put pressure on nurses to merely perform their duties will not reach their main goal of maximizing productivity [4].

Leadership styles have direct effects on the physical and mental health of staff and significant outcomes, such as the improvement of organizational performance, climate, and promotion of job satisfaction [6]. Health-promoting leadership (HPL) refers to the set of behaviors and experiences of leaders that aim to enhance the work environment and staff health [1,7]. HPL promotes health awareness, values, and health-promoting behaviors and plays an crucial role in fostering a healthy work culture in organizations [8], which ensures a reduction in job burnout, job hazards, and an increase in staff attention to the organization's goals, ultimately affecting the improvement of the organization's performance [1]. Indeed, successful HP leaders minimize the dissonance between staff and the environment [9].

The integration between HPL with environmental factors has been more precisely delineated by Jimenez et al. [1]. They identified six areas of work life, including appropriate workload, staff control, reward, values, fairness, and community, along with leaders' awareness of staff' health, as HPL components. Leaders who are cognizant of their staff's health status, participate in related discourses, organize health-promotion programs, and encourage them to participate in these programs have demonstrably healthier staff [1]. It can be inferred that health-promoting leaders instill the understanding that staff are integral to organizations, and that their values are considered congruent with the organization's values [8]. A previous study in Chinese industries during the Covid-19 pandemic shows that HPL, considering the numerous changes that organizations have implemented in response to the health needs of their staff, remains a guide for improving staff relations and promoting work engagement that has been jeopardized during the pandemic [10].

Work engagement is a key characteristic of any organization in the pursuit of success [11]. Staff with poor work engagement often exhibit withdrawal behaviors, which are a source of substantial organizational losses [12]. Work withdrawal behaviors result from staff members' decisions to reduce their work participation. Withdrawal behaviors can manifest in physical and psychological forms: psychological withdrawal primarily encompasses diverse forms such as absurd thoughts, communication, pretending to work, misuse of working time, and web surfing; physical withdrawal refers to the creation of opportunities that staff do for short- or long-term physical escape and includes behaviors such as delays, extended breaks, and, in the most extreme case, turnover [13]. Therefore, it is important to address these costly behaviors and evaluate their underlying components [12]. Recent studies have also indicated that occupational hazards can lead to an escalation in withdrawal behaviors [14]. Few studies of withdrawal behaviors in nursing have elucidated the significant role of organizations. Extended working hours and the resultant stress can ultimately lead to job neglect [15]. One of the strategies employed by effective leaders is to cultivate environments where staff experience a sense of security as they believe that committed staff with high psychological safety will demonstrate superior organizational performance [6]. Psychological safety refers to the conditions under which staff feel secure in expressing novel ideas, challenging existing conditions, and accepting risks within the organization [16]. Nurses experience low psychological safety due to daily exposure to mortality, patients with certain diseases, observation of patients' pain and suffering, psychological burden on patients' companions, and a shortage of organizational resources [17]. According to previous studies, enhancing psychological safety in healthcare systems has yielded beneficial effects such as reducing burnout and creating a reliable environment for consultation and discussing errors [16]. Although recent studies have reported that psychological safety of nurses is only approximately 30%, which is not only a warning sign for the continuation of the high-quality nursing services in society, but also an indicator of inadequate leadership strategies in this profession [18].

In summary, it can be concluded that there exists a bidirectional relationship between staff health and organization, such that not only does the working environment affect the physical and mental health of staff, but staff health also influences the organization and its performance [3]. Withdrawal from an organization and the shortage of specialized and experienced personnel have directed organizational attentions to issues that affect the health and well-being of staff [19]. Withdrawal from an organization is not only an example of the loss of experienced human resources but also of the diminished desire to participate in work environments, which is known as work withdrawal behavior [13]. The maladaptive behavior of leaders and organizations plays a significant role in creating this disconnection [19]. However, no study has evaluated the relationship between HPL and withdrawal behavior in nursing. Conversely, many studies consider psychological safety as the missing link between different leadership styles and positive organizational outcomes [17], and the leadership style of managers is considered one of the factors affecting the psychological safety of healthcare treatment staff in related studies [18]. However, to date, no studies have been conducted on the relationship between HPL and psychological safety in nursing or other fields.

As HPL is a concept based on the creation of work environments that support the physical and mental health of staff, it has the potential to promote psychological safety and reduce uncivil behaviors, such as job withdrawal. Considering the critical role of nurses in promoting and stabilizing public health, as well as the increasing need to recognize and evaluate the characteristics of leaders and effective leadership styles in nursing, further studies on the effect of HPL on withdrawal behaviors and the psychological safety of nurses are essential.

The results of the present study will provide a deeper understanding of HPL and the studied outcomes and draw the attention of nursing managers to nurses' health, which is at great

risk in the current era. The present study was conducted in a nursing system, which is considered to be one of the most complex and challenging human environments. Structural equation modeling (SEM) was used to test the complex relationships between human variables, which met the objectives of this research. This study aimed to investigate the effect of nursing managers' HPL on nurses' withdrawal behaviors and psychological safety.

## Method

This descriptive and analytical study utilizes the SEM approach. Using G*power (version 3.1.9.7), we determined that 325 nurses were required to achieve 0.8 power for effect size of 0.03, type 1 error of 0.05, and two predictors. Considering an expected dropout rate of 10%, 357 participants working in educational hospitals in western provinces were recruited by multi-stage stratified sampling. Using this sampling method, the share of each hospital was determined based on the calculated sample size. Then, in each hospital, the share of each department was also determined, and in each department, a random drawing was conducted from the list of nurses in each department, and self-administered questionnaires were provided to the nurses. Finally, 346 valid questionnaires were used in the final analysis. Data collection was done in 2023.

The inclusion criteria were consent to participate in the study, full-time hospital employment, and a minimum of six months of work experience. The exclusion criteria were incomplete completion of the questionnaires, history of a known mental illness, use of anti-anxiety and depression medications, presence of medical problems and chronic diseases. Questionnaires were administered to participants at the beginning of each shift and collected at the end of each shift. To mitigate bias, non-participation and absence of head nurses during the data collection phase were considered.

## Instruments

Upon completion of the demographic and occupational information form, nurses were instructed to complete the initial questionnaire regarding their perception of their direct manager's behaviors, specifically the head nurse of the ward, followed by two subsequent questionnaires pertaining to themselves.

**Health-promoting leadership questionnaire (HPLQ).** The HPLQ has been developed by Jimenez et al [1]. This questionnaire consists of 21 items and seven components: workload, control, rewards, community, fairness, values, and health awareness, scored on a seven-point Likert Scale ranging from 0(never) to 6(always). The possible range is 0–162, with higher scores indicating that leaders pay more attention to HPL [1].

Example items of this survey include: "my leader takes care that work does not significantly affect private life" (workload); "my leader takes care that the resources and scope for personal development at work can be influenced"(control); "my leader takes care that work is appreciated" (rewards); "my leader take care that work colleagues support each other"(community); "my leader takes care that all resources are fairly distributed" (fairness); "my leader take care that the employees' daily activities correspond with the organization's objectives"(values); and "my leader takes care that all employees are motivated to take care of their health"(Health awareness).

The health-promoting leadership questionnaire was translated and its validity and reliability were assessed by Rashidi et al. [20]. The validity was assessed using confirmatory factor analysis The reliability for all components of this questionnaire was assessed above α = 0.7 using Cronbach's alpha method.

**Work withdrawal behaviors questionnaire.** This questionnaire developed by Hanisch & Hulin [21]. This is a 12-item questionnaire scored on an 8-point Likert scale ranging from

1(never), 2(once a year), 3(two or three times a year), 4(every few months), 5(once a month), 6(more than once a month), 7(once a week), and 8(more than once a week). The possible score range was 12–96, with higher scores indicating more severe withdrawal behavior. In their study, these two researchers reported a Cronbach's alpha of 0.77. Example items of this survey include: "How likely is it that you will resign from your current job in the next several months"; "How often are you late for work" [22].

This questionnaire was translated, and its validity and reliability were assessed by Arshady et al. [23] The validity of this instrument was determined using confirmatory factor analysis and its reliability was assessed using Cronbach's alpha (α = 0.88).

**Edmonson's psychological safety.** This is a 7-item questionnaire scored on a 5-point Likert scale ranging from 1(never) to 5(always). The possible score range was 7–35, with higher scores indicating greater psychological safety. The reliability of this questionnaire was assessed using Cronbach's alpha (α = 0.84). A sample item is: If you make a mistake on your team, is it held against you? [24]. This questionnaire was translated by Hassani& Shohoodi [25], the validity was assessed using confirmatory factor analysis, and its reliability was assessed using Cronbach's alpha (α = 0.8) [26].

## Data analysis method

Quantitative variables were assessed using descriptive statistics and qualitative variables were investigated using frequencies and percentages. SEM establishes the relationship between the measurement model and the structural model based on the assumptions of the conceptual framework. It combines factor analysis and linear regression. Likewise, regression models are additive, whereas SEM is relational in nature, which distinguishes between the regression and SEM decision-making approaches. SEM attempts to justify the acceptance or rejection of a proposed hypothesis by examining the direct and indirect effects of mediators on the relationship between an independent variable and dependent variable [27]. Pearson correlation tests and partial least squares structural equation modeling (PLS-SEM) were used to investigate the relationships between variables. PLS-SEM is a robust statistical methodology extensively employed in the social sciences, including human resource management (HRM) research, to elucidate complex relationships among variables and evaluate sophisticated theoretical frameworks [28]. All statistical tests were performed using SPSS ver.25 and SEM in smart PLS ver.3 software. Statistical significance was set at $P < 0.05$. Data were screened for respondent misconduct (e.g., multiple or extreme responses). The screening was good, and all responses were considered for the analysis. Initially, the data were controlled for participant characteristics because of their probable relationship with dependent variables. The analysis revealed a nonsignificant association; thus, the model was estimated using the main study variables.

## Ethical considerations

In this study, all participants were informed that their involvement was voluntary, and that their privacy was maintained. Prior to completing questionnaires, informed and written consent were obtained. Questionnaires were collected in a single envelope to ensure the confidentiality of nurses' responses and anonymous data collection.

Ethical approval for the study was granted by the Ethics Committee of the Lorestan University of Medical Sciences with the Ethics Code (IR.LUMS.1401.102).

## Results

In total, 346 completed questionnaires were included in the analysis.

The demographic data of the participants are presented in Table 1. The majority of participants were female, married, aged 20–30 years, had 11–20 years of work experience, possessed a bachelor's degree, and were permanent employees.

According to the Pearson test (Table 2), there was a significant relationship between all research variables enabling the evaluation of these variables using SEM. The conceptual model's fit was assessed in two stages: measurement and structural, which are elaborated upon in the subsequent sections.

The standardized regression coefficients of Health-Promoting Leadership (HPL) for psychological safety and withdrawal behaviors were 0.651 and 0.56, respectively. Additionally, the coefficient of determination ($R^2$) of HPL with psychological safety and

**Table 1. Nurses' distribution according to their demographic and work-related characteristics.**

| Variable | Category | No. | Percent (%) | Cumulative percentage (%) |
|---|---|---|---|---|
| Age(years) | 20–30 | 194 | 56.1 | 56.1 |
| | 31–40 | 127 | 36.7 | 92.8 |
| | >40 | 25 | 7.2 | 100 |
| Gender | Male | 67 | 19.4 | 19.4 |
| | Female | 279 | 80.6 | 100 |
| Marriage status | Single | 163 | 47.1 | 47.1 |
| | Married | 183–52.9 | 100 | |
| Years of work experience | 1–10 | 272 | 78.6 | 78.6 |
| | 11–20 | 65 | 18.8 | 97.4 |
| | 21–30 | 9 | 2.6 | 100 |
| Type of employment | Formal | 168 | 48.5 | 48.5 |
| | Contract basis | 40 | 11.6 | 60.1 |
| | Corporate | 25 | 7.2 | 67.3 |
| | Human resources | 113 | 32.7 | 100 |
| Education | B.SC. | 322 | 93.1 | 93.1 |
| | MSc | 24 | 6.9 | 100 |
| Working unit | ICUs | 91 | 26.3 | 26.3 |
| | CCU | 38 | 11 | 37.3 |
| | Surgical | 47 | 13.6 | 50.9 |
| | Medical | 99 | 28.6 | 79.5 |
| | Emergency Department | 38 | 11 | 90.5 |
| | Pediatrics | 33 | 9.5 | 100 |
| Number of shifts | 1–10 | 3 | 0.9 | 0.9 |
| | 11–20 | 136 | 39.2 | 40.1 |
| | 21–30 | 186 | 53.8 | 93.9 |
| | 31–40 | 21 | 6.1 | 100 |

**Table 2. Matrices of correlation coefficients of research variables.**

| Variables | Health promotion leadership | Work withdrawal behaviors | Psychological safety |
|---|---|---|---|
| Health promotion leadership | 1.00 | −0.28[*] | 0.527[*] |
| Work withdrawal behaviors | −0.28[*] | 1.00 | −0.19[*] |
| Psychological safety | 0.527[*] | −0.19[*] | 1.00 |

[*]$p < 0.01$.

withdrawal behaviors was 0.45 and 0.52, respectively. This indicates that, HPL could explain 45% and 52% of the variance in psychological safety and withdrawal behaviors, respectively (Fig 1).

Fig 2 illustrates the significance of each relationship (arrows). A T-value greater than 1.96 indicates a significant relationship. Consequently, it can be concluded that all relationships are significant. Thus, HPL demonstrates an impact on nurses' psychological safety and withdrawal behaviors.

The Cronbach's alpha coefficients, composite reliability, and Average Variance Extracted (AVE) were all within the appropriate ranges, confirming the reliability and convergent validity of the external relations of the research model.

Table 3 presents the fit indices of the research model. Based on the obtained values and in accordance with Kok Wah et al. [27], the data collected for measuring the latent variables demonstrated adequate fit. Therefore, the estimation results of the research model can be considered reliable and valid.

Furthermore, the Q2 (Stone–Geisser) statistic, which represents the predictive fit of the model, yielded values of 0.25, 0.42, and 0.32 for the HPL, psychological safety, and withdrawal behavior variables, respectively. The positive values for these variables indicated an acceptable predictive fit for these constructs. In summary, the results substantiated the conceptual model employed in this study.

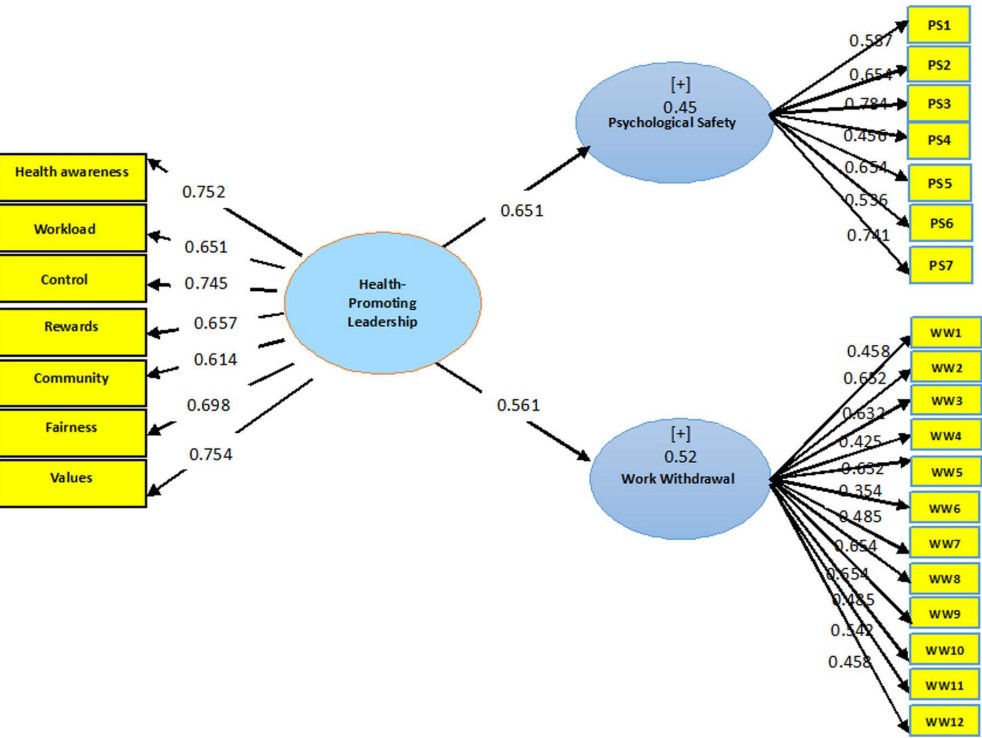

**Fig 1. Conceptual model of standardized coefficients for health-promoting leadership, work withdrawal behaviors, and psychological safety.**

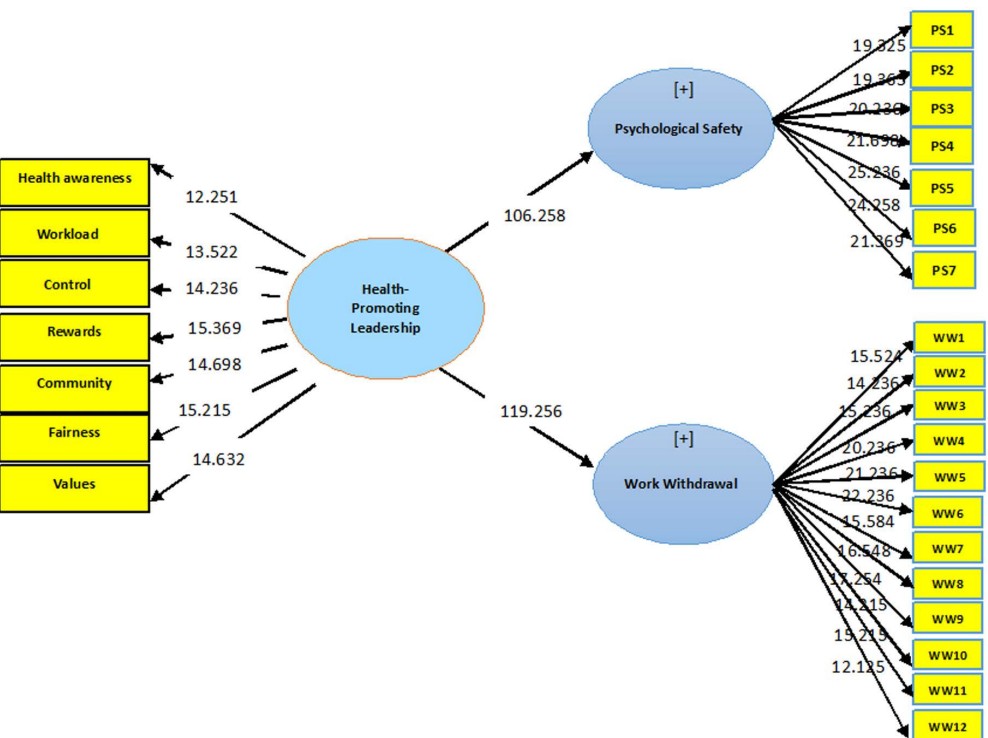

**Fig 2. T-Value values for health-promoting leadership, work withdrawal behaviors, and psychological safety.**

Table 3. Goodness-of-fit statistic tests for hypothesis mode.

| Variables | Acceptable range | Observed value | Result |
|---|---|---|---|
| SRMR [1] | <0.08 | 0.065 | Good fit |
| d-ULS[2] | <0.95 | 0.155 | Good fit |
| d-G[3] | <0.95 | 0.074 | Good fit |
| NFI[4] | >0.50 | 0.856 | Good fit |
| GOF[5] | >0.25 | 0.284 | Good fit |

[1]standardized root mean squared residual,

[2]squared Euclidean distance,

[3]geodesic distance,

[4]Normed Fit Index,

[5]Goodness of Fit.

## Discussion

This study aimed to evaluate the impact of HPL on nurses' psychological safety and withdrawal behavior. A comprehensive literature review revealed that this evaluation has not been previously conducted in nursing. As this is one of the initial nursing studies to evaluate these components, other relevant studies and concepts analogous to the components investigated in this study were utilized to discuss the results. The findings of the present study showed that HPL explained 45% and 52% of the variance in psychological safety and withdrawal behaviors, respectively. To predict human collisions, the values of the coefficients of determination are typically less than 50%. Human behavior is not as readily predicted as physical performance [29].

The results of the present study showed that HPL had an impact on psychological safety, which is aligns with a study on police officers that suggested a negative relationship between health-oriented leadership and health problems (burnout, depression, and physical problems) [9]. Gorgenyi-Hegyes et al. [4] identified the three dimensions of emotional health, physical health, and the provision of health services by the organization as the primary dimensions of a health-promoting environment, which ensures the achievement of outcomes such as employee loyalty. Other studies have shown that HPL aims to meet the basic psychological needs of staff (including a sense of belonging, competence, and autonomy) by enhancing their level of health knowledge and skills, appropriate workload, and job control, which ultimately reduces emotional fatigue [7]. The results of this study also contribute to the promotion of nurses' psychological safety by addressing their psychological needs.

Leaders who foster an organizational climate by promoting creative thinking, face-to-face communication, opportunities to discuss errors, and tolerance of interpersonal risks enhance the psychological safety of the healthcare staff [18]. However, nurses' perceptions of health-promoting leaders pertain to the psychological aspects of the process rather than the physical aspects, such that nurses consider responsibility, promotion of professional capabilities, open communication, and pragmatism as the characteristics of such leaders [30]. The significant effect of safe and healthy working conditions, sense of safety and opportunity for growth, adequate reward, community, work-life harmony on the psychological empowerment of nurses is referenced in the study by Permarupan et al. [31]. They found that these components exhibited a high affinity for the HPL concept. Additionally, the role of establishing open communication, employee participation in decision-making, availability and fair treatment of employee errors, which are among the characteristics of inclusive leadership and are closely related to the principles of HPL, in promoting the psychological safety of nurses are mentioned in Lee et al. [32].

Research on psychological safety, particularly in relation to change-oriented leadership, has been highlighted in several other studies. This leadership style, while emphasizing environmental monitoring, encouraging creative thinking, and envisioning changes, facilitates the improvement of nurses' psychological safety and team performance. The aforementioned characteristics significantly overlap with the HPL concept, which from the nurses' perspective encompasses being alert, promoting career progression, establishing open communication, and being pragmatic [30]. The literature review and results of the current study show a significant close relationship between the factors that contribute to the promotion of psychological safety and the improvement of the mental health of staff based on the HPL concept.

The findings also indicate that HPL can explain 52% of the variance in withdrawal behaviors, which aligns with Karimi et al. [15], who posited that creating work environments characterized by equality, positive leader-staff relationships, encouragement, and workload revision are effective in reducing withdrawal behaviors among nurses. It appears that among the organizational components, leader–member relationships are strongly associated with withdrawal behaviors.

HP leaders facilitate increased job satisfaction and work engagement by providing necessary organizational resources and establishing mutually beneficial relationship with staff [7]. This phenomenon was also observed in a study by Aggarwal et al. [12], who determined that leaders capable of establishing positive bidirectional communication with their staff through the enhancement of staff's psychological capabilities observed lower withdrawal behaviors. A study examining the effect of Machiavellianism on work engagement revealed that managers solely focused on achieving goals through any means, including potentially harming their staff, will not enhance work engagement in contemporary organizations. Consequently, the influence of leadership styles is comprehensible in conjunction with the behavioral

characteristics of leaders, which are associated with the human values encompassed in HPL and Yao's intelligent description, wherein HPL is not just a style but also a set of leadership characteristics [7,33]. However, Liu et al.'s 2021 study revealed that health-promoting leaders serve as a catalyst for promoting work engagement by reducing workload, allocating sufficient organizational resources to each employee, and fostering a sense of value among staff by addressing their health needs. The author references the role of leadership in promoting individual health. In other words, emphasizing personal health values, which are considered an individual's life capital, is effective approach to elicit cooperation from all organizational staff. Another significant organizational factor related to withdrawal behavior is staff's perception of leadership styles. For example, Nauman et al. [34] found that an authoritarian leadership style increases the frequency of withdrawal behavior and decreases job performance. The aforementioned study mentions the role of quality of work life, which is closely associated with the HPL concept, in mitigating the detrimental effects of authoritarian leadership. Leadership style exerts numerous effects on staff personality. Song and Lee [13] found that servant leadership style functioned as a mediating factor between the components of active personality and withdrawal behaviors, indicating that employees with an active personality who benefited from servant leadership exhibited withdrawal behaviors less frequently.

Specifically, support, empowerment, and guidance, which are characteristics of servant leadership and are emphasized in HPL, can contribute to the reduction of withdrawal behaviors by adjusting individual characteristics, which is aligns with the results of the present study. However, the results of the study by Houlihan [8] are inconsistent with those of the current study. The findings indicated that leaders who adopted a democratic leadership style reported experiencing withdrawal behaviors more frequently, whereas those who employed an autocratic leadership style reported these behaviors less frequently. This discrepancy can be attributed to differences in the study populations, cultural differences between developing and developed countries, and data collection instruments. In the aforementioned study, only physical withdrawal behaviors such as lateness or leaving the workplace early and extended breaks were investigated, whereas psychological withdrawal behaviors are more complex and extensive than their physical dimensions. That is, staff are physically present in the organization but do not show proper work efficiency. It can also be inferred that the HPL encompasses a more logical form of democratic leadership. Democratic leadership emphasizes the autonomy of the staff and expression of opinions [34]. Indeed, the higher prevalence of withdrawal behaviors in democratic leadership styles compared to autocratic ones can be attributed to the fact that autocratic leadership styles have been practiced in organizations for a long time. However, to implement styles such as democratic leadership, leaders need to be more goal-oriented and possess a better understanding; this goal orientation has reached maturity in the HPL.

Therefore, it can be concluded that there is an increased necessity for HP leaders due to the complex and challenging nature of working in medical environments, particularly in the nursing profession, as well as the shortage of staff. Indeed, healthcare system personnel incur substantial personal costs to save lives and promote public health, often at the expense of their own well-being. In the meantime, HP leaders not only assist in maintain valuable human resources within the healthcare system but also reduce the frequency of withdrawal behaviors by addressing the psychological needs of staff and providing them with psychological safety, which itself is considered a significant organizational challenge. As there have been few studies on HPL in the nursing field thus far, we hope that the current study will garner the attention of researchers and encourage further investigation in the future.

One of the strengths of the present study is that it is the first in the nursing field; therefore, it can contribute to the formation of a conceptual framework in this domain, particularly regarding how HPL can improve performance and aid in retaining valuable and expert

nursing staff. One of the major limitations of this study was the self-reported evaluation of the components (questionnaire), which could have influenced the results. Additionally, nurses may be reluctant to report withdrawal behaviors as organizational behaviors; therefore, to mitigate this limitation, each participant was provided with a separate envelope to enclose the completed questionnaires.

## Conclusion

The results of the present study revealed the effect of HPL on nurses' withdrawal behavior and psychological safety. As this is one of the initial first nursing studies to evaluate these the above components, subsequent studies can observe and capture staffs' perceptions and subconscious behaviors through interviews and experiments to ensure the accuracy of the data.

### Implications for nursing management

It is anticipated that the results of the present study will draw the attention of nursing managers to HPL and its positive outcomes, and engender a desire to modify leadership styles that lack mutual understanding. Considering the crucial role of nursing managers in enhancing psychological safety and reducing the frequency of withdrawal behaviors, HPL principles can be incorporated into the continuous curriculum review of nursing managers. In addition, incorporation of the HPL principles in the continuous curriculum for nurses without managerial positions can also facilitate the identification of HP leaders, generate demands for this leadership style among nursing leaders, and contribute to the training of future nursing leaders.

## Acknowledgments

This manuscript is extracted form data of research project (the number 2666) approved by Lorestan University of Medical Sciences, Khorramabad, Iran. We wish to thank the Deputy-in-Research of the University, and the nurses who helped us greatly in filling and returning the questionnaires.

## Author contributions

**Conceptualization:** Darya Esmaeilbeigi, Mahdi Sahraei Beiranvand, Fatemeh Mohammadipour.

**Data curation:** Darya Esmaeilbeigi, Mahdi Sahraei Beiranvand, Fatemeh Mohammadipour.

**Formal analysis:** Darya Esmaeilbeigi, Mahdi Sahraei Beiranvand, Fatemeh Mohammadipour.

**Funding acquisition:** Fatemeh Mohammadipour.

**Investigation:** Darya Esmaeilbeigi, Fatemeh Mohammadipour.

**Methodology:** Darya Esmaeilbeigi, Fatemeh Mohammadipour.

**Project administration:** Fatemeh Mohammadipour.

**Software:** Mahdi Sahraei Beiranvand, Fatemeh Mohammadipour.

**Supervision:** Fatemeh Mohammadipour.

**Validation:** Mahdi Sahraei Beiranvand.

**Writing – original draft:** Darya Esmaeilbeigi, Mahdi Sahraei Beiranvand, Fatemeh Mohammadipour.

**Writing – review & editing:** Darya Esmaeilbeigi, Mahdi Sahraei Beiranvand, Fatemeh Mohammadipour.

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
