## [Decision Letter · Decision Letter 0]

2 Jan 2025

PONE-D-24-49553The effect of health-promoting leadership of nursing managers on the

 work withdrawal behaviors and psychological safety of nursesPLOS ONE

Dear Dr. Mohammadipour,

Thank you for submitting your manuscript to PLOS ONE. After careful consideration, we feel that it has merit but does not fully meet PLOS ONE’s publication criteria as it currently stands. Therefore, we invite you to submit a revised version of the manuscript that addresses the points raised during the review process. Please submit your revised manuscript by Feb 16 2025 11:59PM. If you will need more time than this to complete your revisions, please reply to this message or contact the journal office at plosone@plos.org . Please include the following items when submitting your revised manuscript:

We look forward to receiving your revised manuscript.

Kind regards,

Roaa Sabri Gassas

Academic Editor

PLOS ONE

Journal Requirements:

2. In this instance it seems there may be acceptable restrictions in place that prevent the public sharing of your minimal data. However, in line with our goal of ensuring long-term data availability to all interested researchers, PLOS’ Data Policy states that authors cannot be the sole named individuals responsible for ensuring data access (http://journals.plos.org/plosone/s/data-availability#loc-acceptable-data-sharing-methods).

Reviewers' comments:

Reviewer's Responses to Questions

**Comments to the Author**

1. Is the manuscript technically sound, and do the data support the conclusions?

Reviewer #1: Yes

Reviewer #2: Partly

2. Has the statistical analysis been performed appropriately and rigorously? 

Reviewer #1: I Don't Know

Reviewer #2: Yes

3. Have the authors made all data underlying the findings in their manuscript fully available?

Reviewer #1: Yes

Reviewer #2: Yes

4. Is the manuscript presented in an intelligible fashion and written in standard English?

Reviewer #1: Yes

Reviewer #2: No

5. Review Comments to the Author

Reviewer #1: 1.SEM model should be more detailed, such as predecessors if the study of SEM model

2.RESULTS：There should be no such thing as “Zero”. It should be precise to the number of women. The number of women is not shown in Table 1. Please complete.

3.The indicators in the table 2. do not meet the requirements, please check

Reviewer #2: Thank you for the opportunity to review this manuscript. The topic of health-promoting leadership and its association with organizational outcomes is both timely and significant. The study has the potential to contribute meaningfully to the literature; however, I would like to provide some feedback to enhance the clarity, rigor, and interpretability of the manuscript.

General Comments

The manuscript demonstrates substantial effort and commitment to exploring an important area in organizational health research. However, the translation to English has introduced some challenges. Specifically:

1. Language and Terminology:

- The manuscript contains instances of imprecise terminology, phrasing, and redundancy that could potentially mislead readers. For example, terms such as "proven" are not typically used in scientific writing when describing findings from correlational studies. Phrases like "associated with" or "linked to" would be more appropriate in this context.

- The language in the manuscript could benefit from significant refinement to improve readability and avoid ambiguity. I strongly recommend seeking the assistance of a professional English editor familiar with scientific writing to ensure clarity and accuracy.

Specific Comments

1. Study Methodology:

- The manuscript would benefit from additional details about the methodology:

- Effect Size: The choice of an effect size threshold of 0.03. Could the authors elaborate on why this threshold was chosen and how it aligns with the study’s objectives?

- Sampling: It is unclear whether the sample was a convenience sample or selected through other means. Additional information on sampling strategy and its implications for generalizability would strengthen the manuscript.

- Please include a table with descriptive statistics of all study variables including participant demographics.

2. Measurement Instruments:

- The manuscript uses several instruments, but more detail is needed to enhance transparency:

- Health Promoting Leadership Questionnaire: Providing a brief description of this instrument, including sample questions, would help readers better understand the construct being measured. The authors should clarify whether they used the employee-perception component, the leader self-assessment component, or both. Additionally, it would be helpful to specify whom the employees were asked to consider when completing the questionnaire (e.g., head nurse, nurse manager) as different organizations use different nursing leadership structures.

- Work Withdrawal Behaviors Questionnaire: Providing a brief description of this instrument, including sample questions, would help readers better understand the construct being measured.

- Edmonson’s Psychological Safety Questionnaire: Similarly, a brief discussion of this instrument and a few example questions would clarify how psychological safety was operationalized.

3. Findings and Interpretation:

- Some of the conclusions drawn in the manuscript are presented in a causal manner, which is not supported by the study’s design. For example, in the Abstract, the statement: "The results of the present study showed that leaders who put health-promoting leadership on their agenda witnessed positive organizational outcomes, including improved psychological safety and fewer withdrawal behaviors" implies causation which the authors extend to organizational outcomes -- which are not examined as an outcome in this study. As this study appears to be descriptive and correlational, it is more appropriate to describe relationships as associations rather than causal effects. This issue may stem from translation challenges rather than conceptual errors, but it requires correction for accuracy.

Suggestions for Revision

1. Revise language and terminology throughout the manuscript for precision and scientific rigor. Focus on using phrasing consistent with correlational findings.

2. Provide more detail on the methodology, including the rationale for the effect size threshold, sampling strategy, and the use of measurement instruments.

3. Clarify how the Health Promoting Leadership Questionnaire was applied, and specify the target leaders employees were asked to evaluate.

4. Include sample items from the Work Withdrawal Behaviors Questionnaire and Edmonson’s Psychological Safety Questionnaire to enhance understanding of the constructs.

5. Ensure that all findings and conclusions are framed in a way that reflects the study’s correlational nature.

6. Ensure that all findings and conclusions are not extended beyond the study's scope to include outcomes that were not measured (e.g. "positive organizational outcomes,"

Concluding Remarks

This manuscript addresses an important topic and has the potential to contribute valuable insights. I hope the feedback provided here is helpful for the authors in refining their work. I look forward to seeing the revised version and commend the authors for their efforts in conducting and presenting this study.

6. PLOS authors have the option to publish the peer review history of their article (what does this mean? ). If published, this will include your full peer review and any attached files.

**Do you want your identity to be public for this peer review?** For information about this choice, including consent withdrawal, please see our Privacy Policy .

Reviewer #1: **Yes: ** guoguo zhao

Reviewer #2: No

---

## [Author Response · Author response to Decision Letter 1]

29 Jan 2025

Response to Comments from Editor and Reviewers

Dear Editor

The authors would like to thank reviewers and editor for careful review of our manuscript and providing us with their comments and suggestion to improve the quality of the manuscript. The following responses have been prepared to address all of the reviewers’ comments in a point -by-point fashion. Please note that changes are in Track changes or blue marked in the revised manuscript.

Dear Reviewers

Dear Reviewer 1

Comment 1: SEM model should be more detailed, such as predecessors if the study of SEM model Response to Comment 1: Thank you for your comment . Dear reviewer, due to the length of the “introduction part”, explanations related to SEM model were added in the Methods section, Data Analysis, page 10.

Comment 2: RESULTS：There should be no such thing as “Zero”. It should be precise to the number of women. The number of women is not shown in Table 1. Please complete.

Response to Comment 2: Thanks for your careful review. Dear reviewer we did not have a variable for women in Table 1. In fact, it turned out that Table 1, which was related to demographic information, was accidentally deleted. Based on your comment, the table was added back to the revised file.

Comment 3. The indicators in the table 2. do not meet the requirements, please check

Response to Comment 3: We appreciate the reviewer’ insightful comments and helpful suggestions. We rechecked the table and only in the first part, due to translation errors, the number was typed incorrectly, which we corrected.

Dear Reviewer 2

Comment 1. General Comments

The manuscript demonstrates substantial effort and commitment to exploring an important area in organizational health research. However, the translation to English has introduced some challenges. Specifically:

1. Language and Terminology:

- The manuscript contains instances of imprecise terminology, phrasing, and redundancy that could potentially mislead readers. For example, terms such as "proven" are not typically used in scientific writing when describing findings from correlational studies. Phrases like "associated with" or "linked to" would be more appropriate in this context.

- The language in the manuscript could benefit from significant refinement to improve readability and avoid ambiguity. I strongly recommend seeking the assistance of a professional English editor familiar with scientific writing to ensure clarity and accuracy.

Response to Comment 1: Thank you for your comment. Dear reviewer we used the services of translation agencies to translate the article, and a native editing certificate was attached to the article. However, this time a native Persian-speaking professor also edited the article for content, grammar, and academic accuracy, and the changes are visible in the file with a track changes.

Comment 2. Measurement Instruments:

- The manuscript uses several instruments, but more detail is needed to enhance transparency:

- Health Promoting Leadership Questionnaire: Providing a brief description of this instrument, including sample questions, would help readers better understand the construct being measured. The authors should clarify whether they used the employee-perception component, the leader self-assessment component, or both. Additionally, it would be helpful to specify whom the employees were asked to consider when completing the questionnaire (e.g., head nurse, nurse manager) as different organizations use different nursing leadership structures.

- Work Withdrawal Behaviors Questionnaire: Providing a brief description of this instrument, including sample questions, would help readers better understand the construct being measured.

- Edmonson’s Psychological Safety Questionnaire: Similarly, a brief discussion of this instrument and a few example questions would clarify how psychological safety was operationalized.

Response to Comment 2: Thank you for your comment. All items requested by the reviewer were added to the Method section. Please refer to pages 7-9.

3. Findings and Interpretation:

- Some of the conclusions drawn in the manuscript are presented in a causal manner, which is not supported by the study’s design. For example, in the Abstract, the statement: "The results of the present study showed that leaders who put health-promoting leadership on their agenda witnessed positive organizational outcomes, including improved psychological safety and fewer withdrawal behaviors" implies causation which the authors extend to organizational outcomes -- which are not examined as an outcome in this study. As this study appears to be descriptive and correlational, it is more appropriate to describe relationships as associations rather than causal effects. This issue may stem from translation challenges rather than conceptual errors, but it requires correction for accuracy.

Response to Comment 3: We appreciate the reviewer’ insightful comments and helpful suggestions. The abstract was modified based on the suggestions.

Suggestions for Revision

1. Revise language and terminology throughout the manuscript for precision and scientific rigor. Focus on using phrasing consistent with correlational findings.

2. Provide more detail on the methodology, including the rationale for the effect size threshold, sampling strategy, and the use of measurement instruments.

3. Clarify how the Health Promoting Leadership Questionnaire was applied, and specify the target leaders employees were asked to evaluate.

4. Include sample items from the Work Withdrawal Behaviors Questionnaire and Edmonson’s Psychological Safety Questionnaire to enhance understanding of the constructs.

5. Ensure that all findings and conclusions are framed in a way that reflects the study’s correlational nature.

6. Ensure that all findings and conclusions are not extended beyond the study's scope to include outcomes that were not measured (e.g. "positive organizational outcomes,"

Response to Comment: Thank you for your time and effort. We appreciate the reviewer’ insightful comments and helpful suggestions. We have made all changes suggested by the Reviewer.

---

## [Decision Letter · Decision Letter 1]

2 Mar 2025

The effect of health-promoting leadership of nursing managers on the

 work withdrawal behaviors and psychological safety of nurses

PONE-D-24-49553R1

Dear Dr. Fatemeh Mohammadipour

We’re pleased to inform you that your manuscript has been judged scientifically suitable for publication and will be formally accepted for publication once it meets all outstanding technical requirements.

Kind regards,

Roaa Sabri Gassas

Academic Editor

PLOS ONE

Additional Editor Comments (optional):

Reviewers' comments:

Reviewer's Responses to Questions

**Comments to the Author**

1. If the authors have adequately addressed your comments raised in a previous round of review and you feel that this manuscript is now acceptable for publication, you may indicate that here to bypass the “Comments to the Author” section, enter your conflict of interest statement in the “Confidential to Editor” section, and submit your "Accept" recommendation.

Reviewer #1: All comments have been addressed

2. Is the manuscript technically sound, and do the data support the conclusions?

Reviewer #1: Yes

3. Has the statistical analysis been performed appropriately and rigorously? 

Reviewer #1: Yes

4. Have the authors made all data underlying the findings in their manuscript fully available?

Reviewer #1: Yes

5. Is the manuscript presented in an intelligible fashion and written in standard English?

Reviewer #1: Yes

6. Review Comments to the Author

Reviewer #1: This study focuses on the impact of health promotion leadership (HPL-RRB- on nurses' psychological safety and work withdrawal behaviThe the topic selection is closely related to the practical needs of the high-pressure environment in the nursing industry, which has important practical value.The author has responded well to my questions.

7. PLOS authors have the option to publish the peer review history of their article (what does this mean? ). If published, this will include your full peer review and any attached files.

**Do you want your identity to be public for this peer review?** For information about this choice, including consent withdrawal, please see our Privacy Policy .

Reviewer #1: No

---

## [Editor Report · Acceptance letter]

PONE-D-24-49553R1

PLOS ONE

Dear Dr. Mohammadipour,

I'm pleased to inform you that your manuscript has been deemed suitable for publication in PLOS ONE. Congratulations! Your manuscript is now being handed over to our production team.

Kind regards,

on behalf of

Dr. Roaa Sabri Gassas

Academic Editor

PLOS ONE